# The effect of the CONSORT statement on the amount of "unclear" Risk of Bias reporting in Cochrane Systematic Reviews

**Maaike M. Rademaker**[1,2]*, **Geerte G. J. Ramakers**[1], **Adriana L. Smit**[1,2], **Lotty Hooft**[3,4], **Inge Stegeman**[1,2]

**1** Division of Surgical Specialties, Department of Otorhinolaryngology-Head and Neck Surgery, University Medical Center Utrecht, Utrecht, The Netherlands, **2** UMC Utrecht Brain Center, Utrecht, The Netherlands, **3** Cochrane Netherlands, University Medical Center Utrecht, Utrecht University, Utrecht, The Netherlands, **4** Julius Center for Health Sciences and Primary Care, University Medical Center Utrecht, Utrecht University, Utrecht, The Netherlands

* m.m.rademaker-3@umcutrecht.nl

## Abstract

### Background

The Consolidated Standards of Reporting Trials (CONSORT) statement aims to improve clarity and consistency of transparency of reporting in Randomized Controlled Trials (RCTs). The Cochrane Risk of Bias (RoB) tool for RCTs helps authors to judge the RoB. as "low", "high" or "unclear".

### Objective

In this study we aimed to assess whether the implementation and updates of the CONSORT statement influenced the trend of "unclear" RoB scores of RCTs included in Cochrane systematic reviews.

### Methods

All Cochrane reviews published in December to October 2016 were retrieved. The publication year of RCTS included in the reviews were sorted into time frames (≤1995, 1996–2000, 2001–2009 and ≥2010) based on the release- and updates of the CONSORT statement (1996, 2001 and 2010). The association between "unclear" RoB versus "low or high" RoB and the year of publication in different time frames were calculated using a binary logistic regression.

### Results

Data was extracted from 64 Cochrane reviews, with 989 RCTS (6471 items). The logistic regression showed that the odds of RCTs published ≥2010, compared to ≤1995 were more likely not to report an "unclear" RoB for the total data (Odds Ratio (OR) 0.69 (95% Confidence interval: 0.59–0.80)), random sequence generation (OR 0.32 (0.22–0.47), allocation concealment (0.64 (0.43–0.95)) and incomplete outcome data (OR 0.60 (0.39–0.91)).

**Data Availability Statement:** All relevant data are within the manuscript and its Supporting Information files.

**Funding:** The author(s) received no specific funding for this work.

**Competing interests:** The authors have declared that no competing interests exist.

## Conclusion

A slight decrease of "unclear" RoB reporting over time was found. To improve quality of reporting authors are encouraged to adhere to reporting guidelines.

## Introduction

There is increasing concern about the accuracy of the outcomes of biomedical studies and the translatability in clinical care [1]. Inadequate reporting of results leads to imperfect healthcare decisions, that may lead to a waste of health care funds and potential harm to patients [1,2]. The absence of relevant information about the methods of the study, or information about the intervention, and the study results makes replication of studies and implementation in clinical care difficult [3]. In 2014 the Lancet published a series of five articles called '*Increasing value, reducing waste*' [4–8]. This series identified five stages of avoidable waste or inefficiency in biomedical research. One of them is the writing of unbiased and usable research reports [8].

In an effort to improve the quality of research reports, reporting guidelines have been developed. The Consolidated Standards of Reporting Trials (CONSORT) statement (www.consort-statement.org) was developed in 1996 for randomized controlled trials and was updated in 2001 and 2010 [9–12]. Nowadays, many biomedical journals endorse the CONSORT statement [13]. However, Turner et al. showed in 2012 that there are still many flaws in the adherence [14]. For example, where CONSORT provides a flowchart for RCTs, only 263 out of 469 (56%) investigated randomized controlled trials (RCTs) included such a diagram in 2009 [15]. Even the five highest impact general medicine journals show variable and incomplete adherence to the CONSORT statement for abstracts [16].

So far it is unclear whether the publication and updates of the CONSORT statement affected the overall quality and completeness of reporting of RCTs. Standardized quality assessments have been developed to create transparency in the reporting of bias. For randomized controlled trials the most well know is the Cochrane Risk of Bias (RoB) tool introduced in 2008 and updated in 2011 [17]. By this tool, selection bias, performance bias, detection bias, attrition bias and reporting bias are assessed by the judgement of seven items (random sequence generation, allocation concealment, blinding of participants and personnel, blinding of outcome assessment, incomplete outcome data, selective reporting and other bias) as "low", "high" or "unclear" risk of bias [17].

In this study we aim to investigate the effect of the publication and updates of the CONSORT statement on the amount of "unclear" RoB scores of RCTs included in Cochrane systematic reviews. We hypothesize that the CONSORT statement improved completeness of reporting by lowering the amount of "unclear" RoB reporting in systematic reviews.

## Methods

### Selection of Cochrane reviews

We retrieved all therapeutic Cochrane reviews published in December, November and October 2016. We did not make any limitations for topic.

### Data extraction

Data about the risk of bias assessment of all RCTs of included Cochrane systematic reviews were extracted by G.R. The following information from the Cochrane reviews was extracted;

the names and year of publication of the original RCT; the RoB assessments ("low", "high" or "unclear") as judged by the Cochrane review authors. 10% of included reviews was randomly selected by Research Randomizer and accuracy of extracted data by the first screener out of these reviews was randomly checked by M.R [18]. To adhere to the most recent Cochrane handbook (version 5.1) the judgements on the seven key items were extracted (sequence generation, allocation sequence concealment, blinding of participants and personnel, blinding of outcome assessment, incomplete outcome data, selective outcome reporting and other bias). All included Cochrane systematic reviews used the Cochrane RoB tool. However, some reviews did not include all seven key items or used sub-categories for certain types of bias. Data on sub categories was not assessed.

## Data analysis

The frequency of low, unclear and high RoB rating was calculated per item. The year of publication of the individual RCT was related to the publication (1996) and updates (2001, 2010) of the CONSORT statement whereby four time frames were constructed: ≤1995, 1996–2000, 2001–2009 and ≥2010. The proportion of RoB-items scored as unclear, low or high were calculated per time frame, in total and per item. A binary logistic regression analysis was performed. The outcome was RoB assessment "unclear" versus "low and high" RoB scores, per RoB item and in total. The timeframe ≤1995 was the reference timeframe. An odds ratio was deducted (OR, 95% confidence interval (CI)). A binary logistic regression analysis was also performed with the same outcome (RoB "unclear) with time as a continuous variable.

The statistical analyses were performed using SPSS v25.

## Results

### Selection and general characteristics

The search retrieved 75 systematic reviews from the Cochrane Library. 11 were so called "empty reviews" and were excluded. The remaining 64 therapeutic Cochrane systematic reviews included a total of 1008 RCTs. Of these, 19 RCTs were excluded because of several reasons; protocols (n = 5) and no clear publication year (n = 14). Therefore, 989 RCTs (1 double, 988 unique RCTs remaining) were included for further analysis, including a mean of 15.45 (range 1–64) RCTs per review. Fig 1 shows the inclusion process. 10% of the reviews were randomly checked for accuracy of data extraction (n = 7) which encompassed 145 RCTs and 1015 RoB assessments. 19 of 1015 RoB assessments were missings.

Of the 989 included RCTs, 153 RCTs were published in or before 1995 (16%), 94 RCTs in 1996–2000 (10%), 361 RCTs from 2000 to 2009 (37%), and 381 RCTs from 2010 till 2016 (39%). The median publication year was 2008 (interquartile range 12, with a minimum 1967 and maximum of 2016).

### RoB scores of included RCTs

Of the 989 RCTs, random sequence generation was assessed 989 times (100%), allocation concealment 989 times (100%), blinding of participants and personnel 957 times (97%), blinding of outcome assessment 798 times (81%), incomplete outcome data 989 times (100%), selective reporting 955 times (97%) and other bias 794 times (80%) by the authors of the Cochrane systematic reviews. In total 6471 items were assessed, the median number of items per RCT was 7 (IQR 1, minimum 4 –maximum 7).

 

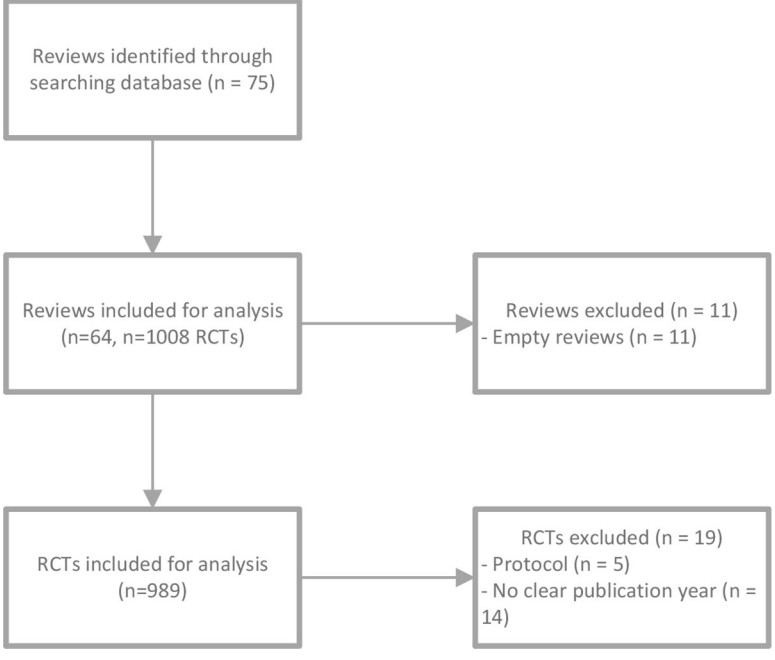

**Fig 1. Flowchart study selection.**

## Assessment of "unclear" RoB reporting

A total of 39% of all items were reported as "unclear", with a maximum of 61% for allocation concealment. The proportion was lower for blinding of participants and personnel (24%) and incomplete outcome data (25%). (Table 1) The reporting of "unclear" RoB over time is illustrated in Table 1 and Fig 2.

## Time

Binary logistic regression shows no statistical significant differences in the total percentage of unclear RoB scores of RCTs between ≤1995, and both 1996–2000 and 2001–2009. However, RCTs in the time frame ≥ 2010, the odds were significantly more likely not to score an "unclear" RoB (OR 0.69 (95%CI 0.59–0.80)) (Table 2)

The odds were significantly more likely not to score an "unclear" RoB in RCTs that were published in the two latest time frames, for the item *random sequence generation* (2001–2009: OR 0.41 (95%CI 0.28–0.61) and ≥ 2010: OR 0.32 (95%CI 0.22–0.47)). For RCTs published in

**Table 1. Numbers and percentages of unclear reporting per time frame of publication of RCT and per Cochrane RoB key item in percentages.**

| Time Frame | Total (n = 6471) | Random sequence generation (n = 989) | Allocation concealment (n = 989) | Blinding of participants and personnel (n = 957) | Blinding of outcome assessment (n = 798) | Incomplete outcome data (n = 989) | Selective reporting (n = 955) | Other bias (n = 794) |
|---|---|---|---|---|---|---|---|---|
| ≤1995 | 443 (44%) | 100 (65%) | 101 (66%) | 34 (24%) | 39 (33%) | 48 (31%) | 68 (45%) | 53 (37%) |
| 1996–2000 | 263 (43%) | 52 (55%) | 60 (64%) | 23 (25%) | 24 (34%) | 19 (20%) | 52 (58%) | 33 (42%) |
| 2001–2009 | 959 (41%) | 158 (44%) | 230 (64%) | 90 (25%) | 100 (34%) | 98 (27%) | 162 (47%) | 121 (42%) |
| ≥ 2010 | 862 (35%) | 143 (38%) | 211 (55%) | 78 (21%) | 86 (27%) | 82 (22%) | 158 (43%) | 104 (36%) |
| Total | 2527 (39%) | 453 (46%) | 602 (61%) | 225 (24%) | 249 (31%) | 247 (25%) | 440 (46%) | 311 (39%) |

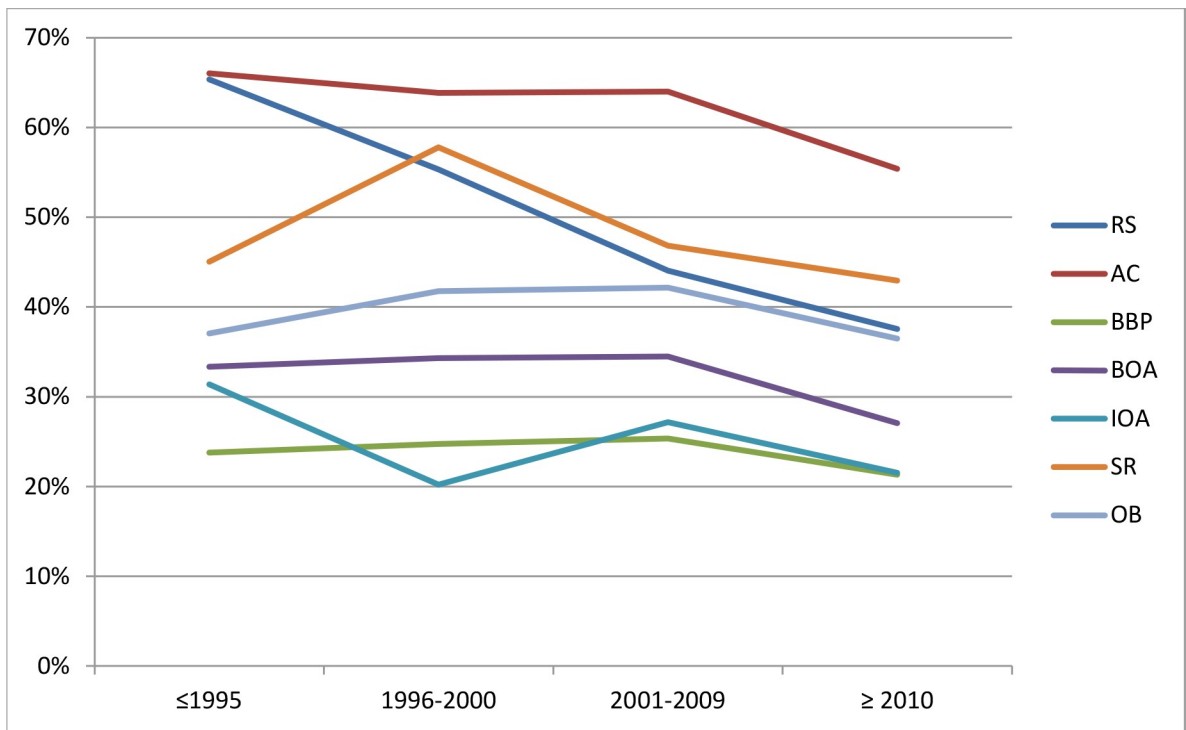

**Fig 2. "Unclear" RoB reporting over time of publication of RCT per Cochrane RoB-item and in total in percentages.** RS = random sequence generation, AC = allocation concealment, BPP = blinding of participants and personnel, BOA = blinding of outcome assessment, IOD = incomplete outcome data, SR = selective reporting, OB = other bias.

time frame ≥ 2010, the odds were significantly more likely not to score an "unclear" Rob for allocation concealment (OR 0.64 (95%CI 0.43–0.95)) and incomplete outcome data (OR 0.60 (95%CI 0.39–0.91)).

With time as a continuous variable, the odds were significantly more likely not to score an unclear risk of bias for all RoB items together (total) (OR 0.987 (0.0981–0.992). (Table 3) The same is applicable for *random sequence generation* (OR: 0.959 (95% CI 0.945–0.973), *allocation concealment* (OR 0.985 (95% CI 0.985 (0.970–0.999) and *incomplete outcome data* (0.982 (95% CI 0.967–0.997).

**Table 2. Binary logistic regression analysis.**

| Time Frame | Total (n = 6471) | Random sequence generation (n = 989) | Allocation concealment (n = 989) | Blinding of participants and personnel (n = 957) | Blinding of outcome assessment (n = 798) | Incomplete outcome data (n = 989) | Selective reporting (n = 955) | Other bias (n = 794) |
|---|---|---|---|---|---|---|---|---|
| ≤1995 | 1.00 | 1.00 | 1.00 | 1.00 | 1.00 | 1.00 | 1.00 | 1.00 |
| 1996–2000 | 0.96 (0.79–1.18) | 0.66 (0.39–1.11) | 0.91 (0.53–1.56) | 1.05 (0.57–1.94) | 1.04 (0.56–1.95) | 0.55 (0.30–1.02) | 1.67 (0.99–2.83) | 1.22 (0.70–2.14) |
| 2001–2009 | 0.88 (0.76–1.02) | **0.41 (0.28–0.61)** | 0.90 (0.61–1.35) | 1.09 (0.69–1.71) | 1.04 (0.66–1.63) | 0.82 (0.54–1.23) | 1.08 (0.73–1.58) | 1.24 (0.82–1.87) |
| ≥2010 | **0.69 (0.59–0.80)** | **0.32 (0.22–0.47)** | **0.64 (0.43–0.95)** | 0.87 (0.55–1.37) | 0.74 (0.47–1.17) | **0.60 (0.39–0.91)** | 0.92 (0.63–1.35) | 0.98 (0.64–1.48) |

Odds ratios with 95% CI for scoring "unclear" RoB per time frame of publication, compared to RCTS published in time frame ≤1995 are presented in total and per item. Time frame ≤1995 was the reference time frame. Values in bold are statistically significant.

**Table 3. Binary logistic regression analysis.**

| Time Frame | Total (n = 6471) | Random sequence generation (n = 989) | Allocation concealment (n = 989) | Blinding of participants and personnel (n = 957) | Blinding of outcome assessment (n = 798) | Incomplete outcome data (n = 989) | Selective reporting (n = 955) | Other bias (n = 794) |
|---|---|---|---|---|---|---|---|---|
| Year | **0.987 (0.981–0.992)** | **0.959 (0.945–0.973)** | **0.985 (0.970–0.999)** | 0.997 (0.981–1.014) | 0.990 (0.974–1.006) | **0.982 (0.967–0.997)** | 0.994 (0.981–1.008) | 0.998 (0.983–1.013) |

Odds ratios with 95% CI for scoring "unclear" RoB with year as a continuous variable. Values in bold are statistically significant.

## Discussion

In this study we provided an overview of the "unclear" RoB reporting of RCTs published between 1967 and 2016 and included in a therapeutic Cochrane systematic review published in October till December 2016. By creating time frames based on the release and updates of the CONSORT statement (1996, 2001, and 2010) the development of "unclear" RoB reporting was calculated over time. In general, a slight decrease of "unclear" RoB reporting over time was found. Over all items the odds were significantly more likely to not score an "unclear" RoB $\geq$ 2010, compared to $\leq$ 1995. For the individual item *random sequence gene*ration, a key factor in creating RCTs, the odds were significantly more likely to not score an "unclear" RoB in two time frames (2001–2009, $\geq$2010). The same applies in the last time frame ($\geq$ 2010) for allocation concealment and incomplete outcome data.

Interestingly, the three items that show significant lower odds of "unclear" RoB $\geq$2010 compared to <1995 (*random sequence generation, allocation concealment and selective reporting*), also show the highest three percentages of unclear reporting over all time frames. Allocation concealment, blinding and method of randomisation might modify effect estimates in RCT's [19].

The outcomes of our study are in line with earlier studies in other research domains. By our own research group the adherence to the CONSORT statement in otorhinolaryngological literature was studied in 2015, demonstrating that the quality of RCTs published in ENT journals was suboptimal [20]. Similar outcomes have been found for publications in diabetes journals, restorative dentistry and anaesthesiology journals [21–23].

Considering the data on time evolution, similarities can be found with Dechartres et al., Reveiz et al. and Peters et al.; a decrease have been demonstrated in "unclear" risk of bias over time, especially in random sequence generation and allocation concealment, and an overall improvement in the reporting quality was found over time [9,24,25]. No comparisons were made with the CONSORT statement. It appears from our study that the first steps have been taken to increase value and reduce waste, especially in reducing waste from incomplete or unusable reports of biomedical research.

There is uncertainty in the term and meaning of "unclear" RoB. One could argue that unclearness of the existence of a risk, implicates a risk. The missing of information in clinical trials could raise a dilemma in the form of two questions to the risk assessors: 1) are we to give them the benefit of the doubt or 2) were the authors strategic in their decision on what items not to describe? [26] Still, it turns out that a large amount of studies with missing information about blinding had in fact performed the blinding [26]. One could argue the suitability to be more conservative with judging missing information [26,27].

There are concerns about the completeness and correctness of the CONSORT statement regarding trial reporting. It is debated that CONSORT does not include all items to properly assess trial quality, such as the lack of a randomization log, the lack of precision in the

intention to treat (ITT) description and the lack of adequate handling of missing data [28–31]. In this paper we have used the CONSORT statement because it is a commonly used tool, that was adopted by many journals [13,26]. A lower quality of reporting, (e.g. higher amount of unclear risk of bias), does not necessarily mean that the outcomes of the reviews were incorrect. In an era of evidence based medicine, however, the lack of possibilities to replicate and reaffirm does introduce questions about the methodology, outcomes and the use of the findings in clinical practice [28].

A number of methodological issues need to be addressed. A large sample size was used of 6471 RoB- items out of 989 therapeutic RCTs out of different medical fields, published in October, November and December 2016. We believe this provides us with an extensive and representative sample. We relied on the judgment of the authors of the included Cochrane systematic reviews for the RoB assessment of the 7 key-items. Variety might exist; one could argue the consistency of the RoB judgements. We assumed all authors worked according to the Cochrane risk of bias Handbook, introduced in 2011 [17]. However there are limitations to the reliability of the Cochrane RoB tool since low agreement rates have been found between Cochrane- and external reviewers [32,33]. Second, one could imagine that review authors might have excluded RCTs because of their inadequate designs, which can affect our outcome. Third, in this paper we use the RoB tool as a measure to assess quality of reporting. This implicates that no statements can be made about adherence to (potentially missing) individual items of the CONSORT statement. Fourth, in this paper only the seven items from the Cochrane RoB tool were included. If authors had adjusted the tool (e.g. added an item), these adjustments and outcomes were not taken in account. This could have led to missing information.

In our study we found that the odds of RCTs published after 2010 were significantly more likely not to score an "unclear" RoB in total. The same applies for several individual items, such as *random sequence generation*, *allocation concealment* and *incomplete outcome data*. These improvements are encouraging. However, the use of the CONSORT statement, and the level of adherence varies greatly amongst RCTs [20,34]. In order to '*Increase value and decrease waste' in* biomedical research a further encouragement to adhere to the CONSORT statement must be established to improve the quality of reporting of RCTs and consequently better patient care.

## Supporting information

**S1 Dataset.**
(XLSX)

## Author Contributions

**Conceptualization:** Geerte G. J. Ramakers, Adriana L. Smit, Inge Stegeman.

**Formal analysis:** Maaike M. Rademaker, Inge Stegeman.

**Investigation:** Geerte G. J. Ramakers.

**Methodology:** Maaike M. Rademaker, Adriana L. Smit, Inge Stegeman.

**Project administration:** Inge Stegeman.

**Supervision:** Adriana L. Smit, Inge Stegeman.

**Writing – original draft:** Maaike M. Rademaker.

**Writing – review & editing:** Geerte G. J. Ramakers, Adriana L. Smit, Lotty Hooft, Inge Stegeman.

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
