## [Editor Report · Decision Letter 0]

9 Dec 2019

PONE-D-19-31501

The effect of the CONSORT statement on the amount of “unclear” Risk of Bias reporting in Cochrane Systematic Reviews

PLOS ONE

Dear Drs Rademaker,

Thank you for submitting your manuscript to PLOS ONE. After careful consideration, we feel that it has merit but does not fully meet PLOS ONE’s publication criteria as it currently stands. Therefore, we invite you to submit a revised version of the manuscript that addresses the points raised during the review process.

We would appreciate receiving your revised manuscript by Jan 23 2020 11:59PM. To enhance the reproducibility of your results, we recommend that if applicable you deposit your laboratory protocols in protocols.io, where a protocol can be assigned its own identifier (DOI) such that it can be cited independently in the future. For instructions see: http://journals.plos.org/plosone/s/submission-guidelines#loc-laboratory-protocols

We look forward to receiving your revised manuscript.

Kind regards,

Vance W. Berger, PhD

Academic Editor

PLOS ONE

Journal Requirements:

Please ensure that your manuscript meets PLOS ONE's style requirements, including those for file naming. The PLOS ONE style templates can be found at http://www.plosone.org/attachments/PLOSOne_formatting_sample_main_body.pdf and http://www.plosone.org/attachments/PLOSOne_formatting_sample_title_authors_affiliations.pdfPlease discuss why you have limited your search to systematic reviews published December, November and October 2016. We noticed you have some minor occurrence(s) of overlapping text with the following previous publication(s), which needs to be addressed:https://doi.org/10.1371/journal.pone.0173358https://doi.org/10.1136/bmjopen-2018-023729https://doi.org/10.1016/j.jclinane.2017.01.017In your revision ensure you cite all your sources (including your own works), and quote or rephrase any duplicated text outside the Methods section. Further consideration is dependent on these concerns being addressed. Please include captions for your Supporting Information files at the end of your manuscript, and update any in-text citations to match accordingly. Please see our Supporting Information guidelines for more information: http://journals.plos.org/plosone/s/supportinxg-information.

Additional Editor Comments (if provided):

You seem to have missed the opportunity to challenge the very notion of "unclear risk of bias". What does that mean, really? "Risk", by its very definition, conveys a lack of certainty. If we lack certainty about risk, then by definition there is risk. Unclear risk means there is a risk of a risk, which is to say, there is a risk. We are far too tolerant of authors who do not provide sufficient information to allow a reader to assess risk. The onus, or burden of proof, is on the authors to rule out risk. If they fail to do so, then they do NOT get the benefit of the doubt. That is the essence of high risk of bias. See:

Berger, VW (2006). ”Missing Data Should Be More Heartily Penalized”, Journal of Clinical Epidemiology 59, 7, 759-760.

Berger, VW (2012). “Conservative Handling of Missing Data”, Contemporary Clinical Trials 33, 460.

Berger, VW (2012). “Conservative Handling of Missing Information”, Journal of Clinical Epidemiology 65, 1237-1238.

Berger, VW (2012). “Internal Validity and the Risk of Bias: A Case for a Comprehensive Review”, Journal of Anesthesia 26, 802-803.

Berger, VW, Mickenautsch, S (2015). “On the Need for Objective Measures of Risk of Bias”, Contemporary Clinical Trials 41, 202-203.

Also, since you mention CONSORT so prominently, it seems fair to question what it offers, in terms of value added, since 1) it is grossly incomplete; 2) in some cases it is demonstrably wrong; and 3) even where it is correct all it does is repackage sage advice that was already out there prior to 1996. Adherence to CONSORT is neither necessary nor sufficient for a good trial, and you have the opportunity to mention that, too. See:

Palys, K, Berger, VW (2013). “On the Incompleteness of CONSORT”, JNCI 105, 3, 244.

---

## [Author Response · Author response to Decision Letter 0]

23 Jan 2020

Dear dr. Berger,

We would like to thank you for your review of our work. We have read your papers with joy and interest , and have added the mentioned discussion points to the paper. 

Please read our response to the points raised in the letter below.

1. Please ensure that your manuscript meets PLOS ONE's style requirements, including those for file naming

a. We have revised the manuscript to adhere to the guidelines, we have substantially shortened the abstract

2. Please discuss why you have limited your search to systematic reviews published December, November and October 2016.

a. In order to make our work feasible, we have limited our search to December, November and October 2016, we think this provides us with an extensive and representative sample of reviews. 

3. We noticed you have some minor occurrence(s) of overlapping text with the following previous publication(s), which needs to be addressed:

https://doi.org/10.1371/journal.pone.0173358

https://doi.org/10.1136/bmjopen-2018-023729

https://doi.org/10.1016/j.jclinane.2017.01.017

In your revision ensure you cite all your sources (including your own works), and quote or rephrase any duplicated text outside the Methods section. Further consideration is dependent on these concerns being addressed.

a. We did not see the mentioned publications before, and after extensive reading, are unable to find the sentences in our paper. Could you give us an indication of the exact sentences?. 

4. Please include captions for your Supporting Information files at the end of your manuscript, and update any in-text citations to match accordingly. Please see our Supporting Information guidelines for more information: http://journals.plos.org/plosone/s/supportinxg-information.

a. We have included the captions for our supporting information file at the end of the manuscript. 

5. You seem to have missed the opportunity to challenge the very notion of "unclear risk of bias". What does that mean, really? "Risk", by its very definition, conveys a lack of certainty. If we lack certainty about risk, then by definition there is risk. Unclear risk means there is a risk of a risk, which is to say, there is a risk. We are far too tolerant of authors who do not provide sufficient information to allow a reader to assess risk. The onus, or burden of proof, is on the authors to rule out risk. If they fail to do so, then they do NOT get the benefit of the doubt. That is the essence of high risk of bias. See:

a. Thank you very much for your critical note. have now added this point in the discussion section of our paper. 

6. Also, since you mention CONSORT so prominently, it seems fair to question what it offers, in terms of value added, since 1) it is grossly incomplete; 2) in some cases it is demonstrably wrong; and 3) even where it is correct all it does is repackage sage advice that was already out there prior to 1996. Adherence to CONSORT is neither necessary nor sufficient for a good trial, and you have the opportunity to mention that, too. See:

a. We have revised the discussion in concern to this point.

We would like to thank you again for your thorough work. We hope you consider the revisions suitable. We hope you consider our manuscript suitable for peer-review. 

Kind regards,

On behalf of all co-authors,

Maaike Rademaker

---

## [Decision Letter · Decision Letter 1]

3 Mar 2020

PONE-D-19-31501R1

The effect of the CONSORT statement on the amount of “unclear” Risk of Bias reporting in Cochrane Systematic Reviews

PLOS ONE

Dear Drs Rademaker,

Thank you for submitting your manuscript to PLOS ONE. After careful consideration, we feel that it has merit but does not fully meet PLOS ONE’s publication criteria as it currently stands. Therefore, we invite you to submit a revised version of the manuscript that addresses the points raised during the review process.

We would appreciate receiving your revised manuscript by Apr 17 2020 11:59PM. To enhance the reproducibility of your results, we recommend that if applicable you deposit your laboratory protocols in protocols.io, where a protocol can be assigned its own identifier (DOI) such that it can be cited independently in the future. For instructions see: http://journals.plos.org/plosone/s/submission-guidelines#loc-laboratory-protocols

We look forward to receiving your revised manuscript.

Kind regards,

Vance W. Berger, PhD

Academic Editor

PLOS ONE

Additional Editor Comments (if provided):

The revision looks much improved. Please address the reviewer comments, and also please note that the conclusions in the abstract are not, in fact, conclusions, since they are statements of what was directly observed. So this makes them results, and the question remains, what do you conclude from this study?

Reviewers' comments:

Reviewer's Responses to Questions

**Comments to the Author**

1. If the authors have adequately addressed your comments raised in a previous round of review and you feel that this manuscript is now acceptable for publication, you may indicate that here to bypass the “Comments to the Author” section, enter your conflict of interest statement in the “Confidential to Editor” section, and submit your "Accept" recommendation.

Reviewer #1: (No Response)

2. Is the manuscript technically sound, and do the data support the conclusions?

Reviewer #1: Partly

3. Has the statistical analysis been performed appropriately and rigorously? 

Reviewer #1: Yes

4. Have the authors made all data underlying the findings in their manuscript fully available?

Reviewer #1: Yes

5. Is the manuscript presented in an intelligible fashion and written in standard English?

Reviewer #1: Yes

6. Review Comments to the Author

Reviewer #1: This is a relatively straightforward manuscript describing trends in time in the risk of bias in randomized clinical trials as determined by authors of Cochrane systematic reviews. The aim of the study was to evaluate the impact of the CONSORT reporting guidelines on the risk of bias as reported in these trials. The study was completed adequately, but there are a few questions about the methods that should be clarified in the manuscript.

1. Some justification needs to be included in the manuscript for why only the reviews published in December, November, and December 2016 were included. I see the authors responded to this question previously, but the reason needs to be included in the manuscript.

2. Were the 989 RCTs unique. Sometimes RCTs are used in more than one review.

3. Only one person extracted data. It would have been preferably for a second person to have extracted at least a sample. This is especially concerning in that the methods state that “Data on sub categories was not assessed.” Often the risk of bias for blinding/masking is assessed by outcome. If a systematic reviewer assessed risk of bias for more than one outcome, how was the data extracted? Were all these ignored or was there some kind of averaging done?

7. PLOS authors have the option to publish the peer review history of their article (what does this mean?). If published, this will include your full peer review and any attached files.

Reviewer #1: No

---

## [Author Response · Author response to Decision Letter 1]

16 Apr 2020

Dear dr. Berger and reviewer,

Thank you in advance for the effort of reviewing our manuscript. We have revised it carefully and adjusted some items.

Please find our response to the commentary below.

Additional Editor Comments (if provided):

1. The revision looks much improved. Please address the reviewer comments, and also please note that the conclusions in the abstract are not, in fact, conclusions, since they are statements of what was directly observed. So this makes them results, and the question remains, what do you conclude from this study?

We have updated the abstract. 

Reviewer #1: This is a relatively straightforward manuscript describing trends in time in the risk of bias in randomized clinical trials as determined by authors of Cochrane systematic reviews. The aim of the study was to evaluate the impact of the CONSORT reporting guidelines on the risk of bias as reported in these trials. The study was completed adequately, but there are a few questions about the methods that should be clarified in the manuscript.

1. Some justification needs to be included in the manuscript for why only the reviews published in December, November, and December 2016 were included. I see the authors responded to this question previously, but the reason needs to be included in the manuscript.

We have added this to the discussion. 

2. Were the 989 RCTs unique. Sometimes RCTs are used in more than one review.

Thank you for your point. We have checked this and adjusted in the review. There are 988 unique RCTS, one RCT was used for two reviews. Please see line 103. For the record both reviews judged the RoB for this RCT differently. 

3. Only one person extracted data. It would have been preferably for a second person to have extracted at least a sample. 

Thank you for your comment, we have checked 10% of the RCTs randomly with research randomizer.com. This resulted in 145 RCTs and 1015 RoB assesments, of which 19 missings. We were able to discover 4 mistakes data extraction for one RCT (0.39%). We corrected the data file, reran our analyses and updated the result section of the paper. 

This is especially concerning in that the methods state that “Data on sub categories was not assessed.” Often the risk of bias for blinding/masking is assessed by outcome. If a systematic reviewer assessed risk of bias for more than one outcome, how was the data extracted? Were all these ignored or was there some kind of averaging done?

In this paper we stuck to the seven RoB items as defined by the Cochrane Risk of Bias tool. If a review adjusted the tool; we ignored the adjustments and stuck to the ‘pure’ seven RoB items. There was no averaging done. This was also added to the discussion. 

We would like to thank you both again for your revisions. We hope you consider the revisions suitable. 

Kind regards,

On behalf of all co-authors,

Maaike Rademaker

---

## [Decision Letter · Decision Letter 2]

18 Jun 2020

The effect of the CONSORT statement on the amount of “unclear” Risk of Bias reporting in Cochrane Systematic Reviews

PONE-D-19-31501R2

Dear Dr. Rademaker,

We’re pleased to inform you that your manuscript has been judged scientifically suitable for publication and will be formally accepted for publication once it meets all outstanding technical requirements.

Kind regards,

Vance Berger

Academic Editor

PLOS ONE

Additional Editor Comments (optional):

Reviewers' comments:

Reviewer's Responses to Questions

**Comments to the Author**

1. If the authors have adequately addressed your comments raised in a previous round of review and you feel that this manuscript is now acceptable for publication, you may indicate that here to bypass the “Comments to the Author” section, enter your conflict of interest statement in the “Confidential to Editor” section, and submit your "Accept" recommendation.

Reviewer #1: All comments have been addressed

2. Is the manuscript technically sound, and do the data support the conclusions?

Reviewer #1: Yes

3. Has the statistical analysis been performed appropriately and rigorously? 

Reviewer #1: Yes

4. Have the authors made all data underlying the findings in their manuscript fully available?

Reviewer #1: Yes

5. Is the manuscript presented in an intelligible fashion and written in standard English?

Reviewer #1: Yes

6. Review Comments to the Author

Reviewer #1: (No Response)

7. PLOS authors have the option to publish the peer review history of their article (what does this mean?). If published, this will include your full peer review and any attached files.

Reviewer #1: No

---

## [Editor Report · Acceptance letter]

24 Jun 2020

PONE-D-19-31501R2 

The effect of the CONSORT statement on the amount of “unclear” Risk of Bias reporting in Cochrane Systematic Reviews 

Dear Dr. Rademaker:

I'm pleased to inform you that your manuscript has been deemed suitable for publication in PLOS ONE. Congratulations! Your manuscript is now with our production department. 

Kind regards, 

on behalf of

Dr Vance Berger 

Academic Editor

PLOS ONE